# Panobinostat Attenuates Experimental Autoimmune Encephalomyelitis in Mice via Suppressing Oxidative Stress-Related Neuroinflammation and Mitochondrial Dysfunction

**DOI:** 10.3390/ijms252212035

**Published:** 2024-11-09

**Authors:** Yanjia Shen, Jiaying Zhao, Ran Yang, Huilin Yang, Minmin Guo, Baixi Ji, Guanhua Du, Li Li

**Affiliations:** Beijing Key Laboratory of Drug Targets Identification and Drug Screening, Institute of Materia Medica, Chinese Academy of Medical Sciences and Peking Union Medical College, Beijing 100050, China; shenyanjia@kmmu.edu.cn (Y.S.); zhaojiaying1996@126.com (J.Z.); yangimm0123@163.com (R.Y.); huilinyang@imm.ac.cn (H.Y.); guominmin@imm.ac.cn (M.G.); baj34@pitt.edu (B.J.); dugh@imm.ac.cn (G.D.)

**Keywords:** panobinostat, multiple sclerosis, experimental autoimmune encephalomyelitis, neuroinflammation, mitochondrial dysfunction, oxidative stress

## Abstract

Multiple sclerosis (MS) is an autoimmune disease mediated by T helper cells, which is characterized by neuroinflammation, axonal or neuronal loss, demyelination, and astrocytic gliosis. Histone deacetylase inhibitors (HDACis) are noted for their roles in easing inflammatory conditions and suppressing the immune response. Panobinostat, an HDACi, is now being used in treating multiple myeloma. Nevertheless, the effect of panobinostat on autoimmune diseases remains largely unclear. Thus, our research endeavored to determine if the administration of panobinostat could prevent experimental autoimmune encephalomyelitis (EAE) in mice, one of the most commonly used animal models of MS, and further explored the underlying mechanisms. The EAE mice were generated and then administered continuously with panobinostat at a dosage of 30 mg/kg for 16 days. The results indicated that panobinostat markedly alleviated the clinical symptoms of EAE mice, inhibiting demyelination and loss of oligodendrocytes in the central nervous system (CNS). Moreover, panobinostat decreased inflammation and the activation of microglia and astrocytes in the spinal cords of EAE mice. Mechanistically, treatment with panobinosat significantly suppressed M1 microglial polarization by blocking the activation of toll-like receptor 2 (TLR2)/myeloid differentiation factor 88 (MyD88)/interferon regulatory factor 5 (IRF5) pathway. Additionally, panobinostat inhibited mitochondrial dysfunction and reduced oxidative stress in the spinal cords of EAE mice. In conclusion, our findings reveal that panobinostat significantly ameliorates experimental autoimmune encephalomyelitis in mice by inhibiting oxidative stress-linked neuroinflammation and mitochondrial dysfunction.

## 1. Introduction

Multiple sclerosis (MS), a central nervous system (CNS) disabling disease, is characterized by the loss of myelin sheaths, neuro-axonal degeneration, microglia/astrocytes activation, and so on. As a neurological disease that typically leads to disability among young adults, affects 33 individuals per 100,000 worldwide [1]. MS is categorized into four specific subtypes according to the course of the disease in clinical settings: (a) relapsing/remitting multiple sclerosis (RRMS), the predominant form, impacting 85% of individuals with MS, which is marked by cycles of exacerbation followed by intervals of recovery; (b) secondary progressive multiple sclerosis (SPMS), which emerges gradually after the initial start of RRMS; (c) primary progressive multiple sclerosis (PPMS), characterized by a steady and persistent deterioration of neurological performance; and (d) progressive relapsing multiple sclerosis (PRMS), the infrequent subtype (under 5%), similar to PPMS, but with occasional episodes of flare-ups [2]. At the onset of MS, the disease is mainly linked to inflammation and the stimulation of microglia via oxidative stress. However, in the progressive stage, other amplifying mechanisms related to age and lesion burden, for instance, ongoing damage to mitochondria or the buildup of iron in the brain, together with the iron released within areas of demyelination, seem to gain increasing significance [3].

Microglia are similar to mononuclear macrophages present within the CNS, which are located within the brain and spinal cord. They can present antigens and secrete a variety of cytokines, phagocytose pathogens and necrotic tissues. In a condition of equilibrium, microglia are referred to as “resting” microglia, tasked with monitoring the microenvironments. Upon activation, microglia experience significant morphological transformations and differentiate into two distinct phenotypes. Activated M1-type microglia secrete a plethora of inflammatory cytokines, including factors such as monocyte chemoattractant protein-1 (MCP-1), interleukin-6 (IL-6), and tumor necrosis factor α (TNF-α), to promote neuroinflammation. Conversely, M2-type microglia produce substantial quantities of anti-inflammatory factors such as IL-10 and Arginase 1 (Arg1) [4,5]. Neuropathological studies have found there are lymphocytes, microglia, and myelin-laden macrophages in the brain white matter of MS patients, accompanied by different stages of inflammation [6,7]. Studies in animal experiments have shown that decreasing the quantity of M1-type microglia or inhibiting the release of inflammatory factors from microglia could considerably reduce the severity of disease in mice with experimental autoimmune encephalomyelitis (EAE) [8]. Furthermore, the activation of microglia may occur prior to T cell infiltration and demyelination, exacerbating axon damage [9]. Therefore, the early activation of M1-type microglia is a sign of the inflammatory response of the nervous system, and plays a crucial role in the neuroinflammatory process of MS. Consequently, focusing on the interventions that suppress M1-type microglia might offer a hopeful approach for the future MS treatments.

Recent evidence indicates that mitochondrial impairment among individuals with MS is a crucial factor in the damage and loss of both axons and neurons. In a study of function in 10 post-mortem brains of patients with MS, there was a decrease in nuclear DNA encoded mitochondrial transcripts, resulting in insufficiencies within the complexes I and III of the mitochondrial electron transport chain [10]. Another study of 13 patients with MS indicated large mitochondrial DNA (mtDNA) deletions in neurons and a significant decrease in submits of mitochondrial respiratory chain complex IV [11]. In an animal model of MS, double-strand breaks in mtDNA have been observed [12]. Along with deficiencies in mitochondrial respiratory chain complexes and mutations in mtDNA, abnormalities in mitochondrial transport have also been noted in MS. The deficits in mitochondrial trafficking were confirmed in EAE [13]. Further studies have indicated that motor proteins, including kinesin superfamily proteins (KIF5A, KIF21B, and KIF1B), were significantly reduced in progressive MS [14]. Mitigating mitochondrial dysfunction in neuronal compartments could present a promising therapeutic strategy for improving neuronal function in patients with progressive MS.

Growing evidence indicates that oxidative stress is a key factor in the development and progression of MS, which has been heavily linked to pathological characteristics of the disease, including myelin destruction, axonal degeneration, and inflammation [15]. In the brains of patients with MS, oxidative stress has an effect on demyelination and neurodegeneration through directly oxidizing proteins, lipids, and DNA [16]. The destruction of the anti-oxidative defense system has been found in active MS lesions, including the up-regulation of heme oxygenase-1 (HO-1), superoxide dismutase, catalase, and glutathione. Their expression is mainly regulated by PPAR gamma coactivator 1-alpha (PGC-1α) and nuclear factor (erythroid-derived 2)-like 2 (Nrf2). As MS advances, oxidative stress is heavily associated with inflammation and compromised mitochondrial function. In RRMS patients, oxidative stress is involved in the initiation of inflammation in the acute phase of MS [17]. The surge in oxidative activity associated with inflammation in activated microglia is crucial in the process of demyelination. Furthermore, mitochondria are highly vulnerable to oxidative damage. Oxidative injury participates in not only the oxidation of respiratory chain components but also the mutations and deletions of mtDNA. Several studies on the MS EAE model have found that oxidative stress-mediated mitochondrial dysfunction was critical in cell apoptosis and neurodegeneration [18].

Histone deacetylases (HDACs) are widely involved in the pathological process of many diseases, and histone deacetylase inhibitors (HDACis) have been reported to have neuroprotective, anti-inflammatory, and immunomodulatory effects. The immunoregulatory properties of HDACi have been reported to target pathological inflammation in some diseases such as MS, rheumatoid arthritis, and colitis. Panobinostat, a highly potent inhibitor targeting multiple HDACs, has been used clinically for treating multiple myeloma [19]; it has been noted to influence the responses from both the adaptive and innate immune systems in various ways. In human immunodeficiency virus-1 (HIV-1)-infected individuals, panobinostat significantly influenced T cell activation, increased proportions of regulatory T(Treg) cells, and decreased functional mitogen responsiveness [20]. Nevertheless, the effect of panobinostat on autoimmune diseases such as MS is still not well understood. In the current study, we evaluated the effects of the HDACi panobinostat on EAE in mice and investigated the underlying mechanisms.

## 2. Results

### 2.1. Panobinostat Alleviated EAE Symptoms in Mice

To determine the therapeutic potential of panobinostat on MS, panobinostat (30 mg/kg) was administrated daily in a classical myelin oligodendrocyte glycoprotein (MOG)-induced EAE mouse model, beginning on day 13 after immunization. Clinical scores for EAE were continuously tracked for 28-day duration. The neuromotor scores were recorded on day 28 post-immunization, and the suspension behavior score was assessed on days 7, 14, 21, and 28 post-immunization. The mice were euthanized in a humane manner on the 28th day after immunization (Figure 1A). As illustrated in Figure 1B, EAE mice treated with the panobinostat experienced a lesser degree of body weight loss compared to the sodium carboxymethyl cellulose (CMC-Na)-administrated EAE mice throughout the disease’s course. The clinical score in the group treated with 30 mg/kg of panobinostat showed a significant reduction compared to the EAE group starting on day 20 and continuing through to the end of the study (Figure 1C). Additionally, the neuromotor score demonstrated that as compared with the CMC-Na-administrated EAE mice, panobinostat (30 mg/kg) treatment remarkably increased the neuromotor score of EAE mice (Figure 1D). As expected, the suspension behavior scores during EAE progression showed that panobinostat (30 mg/kg) significantly ameliorated the motor dysfunction of EAE mice (Figure 1E). These results suggest that panobinostat has a symptom-relieving effect on EAE mice.

### 2.2. Panobinostat Ameliorated the CNS Demyelination and Loss of Oligodendrocytes in the Spinal Cords of EAE Mice

To assess the impact of panobinostat on CNS pathology in EAE mice, demyelination changes in the spinal cords were detected using Luxol Fast Blue (LFB) (Sigma-Aldrich, Saint Louis, MO, USA) staining. Histological examination of spinal cord samples revealed that, in contrast to the characteristic demyelination associated with the CNS in mice with experimental autoimmune encephalomyelitis, panobinostat (30 mg/kg) administration remarkably inhibited demyelination (Figure 2).

We conducted a deeper investigation into the alterations of myelinating oligodendrocytes within this demyelinating model. Myelin basic protein (MBP) was shown to be a marker for the presence of myelinating oligodendrocytes. Being consistent with the LFB staining results, the area of MBP+ cells in EAE mice decreased compared with the control group, while panobinostat-treated mice showed more myelinating oligodendrocytes surrounding the vascellum and edge of the spinal cord (Figure 2). The findings revealed that treatment with panobinostat safeguards mice with EAE against the loss of myelin sheaths.

### 2.3. Panobinostat Diminished Inflammation and the Microglial/Astrogial Activation in the Spinal Cord Tissues of Mice with EAE

Hematoxylin-eosin (H&E) staining was used to assess the amount of inflammatory cells infiltrating the lumbar area of the spinal cord and to determine whether the cells were present. The results indicated that, in comparison to the vehicle treatment, administration of panobinostat (30 mg/kg) significantly reduced inflammatory cell infiltration in the affected lumbar spinal cords of the EAE mice. In addition to the immune factors involved in the pathogenesis of EAE, activated glial cells also play a crucial role in neurodegenerative processes. We employed the Iba1 marker to examine myeloid cell reactions in the context of EAE. In the control group, Iba1+ cells displayed a broad and consistent distribution throughout the spinal cord parenchyma (Figure 3). These cells were small and branched, characteristic of the morphology typically seen in resting microglia. Within the spinal cord tissues of mice with EAE, there was a large enlargement of the Iba1-positive area surrounding zones where immune cell invasion and loss of myelin sheaths occurred. Panobinostat (30 mg/kg) has been demonstrated to reduce the Iba1+ staining area, resulting in decreased microglial activation and/or infiltration of monocytes/macrophages.

Astrocyte activation in the spinal cord was triggered by EAE, as evidenced by the rise in the percentage of GFAP+ stained area (Figure 3). A notable decrease in the proportion of GFAP-positive areas was observed in mice treated with panobinostat as opposed to those with EAE alone, indicating a lessening of astrogliosis.

### 2.4. Panobinostat Reduced the mRNA Level of Pro-Inflammatory Cytokines and Suppressed M1 Microglia Polarization in the Spinal Cord Tissues of EAE Mice

To elucidate the mechanism of panobinostat on EAE, we next examined how panobinostat impacts the generation of pro-inflammatory cytokines, such as TNF-α, IL-1β and MCP-1. As indicated in Figure 4A–C, the transcriptional levels of TNF-α, IL-1β, and MCP-1 were upregulated significantly in the spinal cords of EAE Mice and panobinostat treatment down-regulated the levels of these pro-inflammatory factors. Following CNS lesions, microglia/macrophages polarize to either the M1 or M2 phenotypes at different stages, with each phenotype having different roles in the repair process. M1 populations are characterized by the expression of signature proteins, such as CD16, CD68, and CD86. To systematically study the effect of panobinostat on microglial polarization, RT-PCR was applied to evaluate the mRNA expression of M1 markers (CD16, CD68, and CD86). Figure 4 illustrates that panobinostat reduced the expression of marker genes in M1-type microglia, indicating that it may have anti-inflammatory effects by inhibiting the formation of M1 microglia.

### 2.5. Panobinostat Inhibited Neuroinflammation via TLR2/MyD88/IRF5 Signaling in the Spinal Cords of EAE Mice

To further explore the potential anti-inflammatory mechanisms of panobinostat in vivo, we examined the toll-like receptor 2 (TLR2)/myeloid differentiation factor 88 (MyD88)/interferon regulatory factor 5 (IRF5) pathway, which is involved in the regulation of neuroinflammation. As shown in Figure 5, compared with the control group, the protein levels of TLR2, MyD88, and IRF5 were markedly enhanced in the spinal cords of EAE mice. Following the intervention with panobinostat, there was a considerable decrease in the amounts of TLR2, MyD88, and IRF5 protein levels.

### 2.6. Panobinostat Mitigated the Impairment of Mitochondrial Function Within the Spinal Cord Tissues of Mice Affected by EAE

As the dynamin-related protein, optic atrophy 1 (OPA1) resides within the inner layer of the mitochondria’s double membrane, whereas mitofusin 2 is situated within the outer membrane. Both of them are crucial for mitochondrial fusion [21]. The down-regulation of mitofusion 2 and OPA1 indicates that there are obstacles in the process of mitochondrial protein fusion. When compared to the control group, there was a significant decrease in the expression levels of OPA1 and mitofusion2 in the EAE group, as shown in Figure 6. However, panobinostat appeared to alleviate this reduction in both proteins, suggesting that it may help restore mitochondrial fusion function in EAE mice. Also, the key protein in the process of mitochondrial fission Drp1 was significantly down-regulated in EAE mice, while panobinostat treatment ameliorated the down-regulation of Drp1, suggesting that panobinostat may ameliorate the dysfunction associated with the mitochondrial fission process.

### 2.7. Panobinostat Mitigated Oxidative Stress Within the Spinal Cord Tissues of Mice with EAE

Oxidative stress is important for the development of MS. NADPH oxidase 2 (NOX2), also known as Gp91phox, plays an important role in the production of NADPH-stimulated superoxide and contributes to oxidative stress during various pathological processes [22]. On the contrary, HO-1 and Nrf2 are two anti-oxidative proteins that act against oxidative stress [23]. As illustrated in Figure 7, the EAE model group exhibited a significant increase in the expression levels of pro-oxidative stress proteins NOX2 and HO-1 compared to the control group, while Nrf2 expression was markedly decreased. However, treatment with panobinostat led to a decrease in NOX2 and HO-1 expression and an increase in Nrf2 expression in the spinal cords of EAE mice.

## 3. Discussion

Multiple sclerosis is a T cell-mediated chronic inflammatory demyelinating disease within the CNS. HDACs are a group of enzymes that deacetylate not only the lysine residues of histones but also numerous proteins with key roles in cell metabolism, signaling, and death. There are five HDACis (including romidepsin, vorinostat, belinostat, tucidinostat, and panobinostat) that have been approved by the US Food and Drug Administration (FDA). Vorinostat has been shown to inhibit the differentiation, maturation, and endocytosis of human CD14+ monocyte-derived dendritic cells, and to alleviate EAE [24]. Our previous study demonstrated that another HDACi belinostat significantly ameliorates EAE through inhibition of the TLR2-mediated MyD88-dependent signaling pathway and elevation of acetylated NF-κB p65 levels via down-regulating HDAC3 [25]. Panobinostat (LBH589), a novel inhibitor of class I, II, and IV HDACs, demonstrates a potency more than ten times greater than vorinostat in inhibiting HDACs, positioning it as potentially the most effective inhibitor among broad-spectrum HDACis [26]. In the present study, our data showed that panobinostat effectively improved the disease condition of EAE mice. Panobinostat is an approved medication that has established clinical experience and safety data, indicating that non-selective HDAC inhibition may be adequate for clinical development in MS.

Toll-like receptors (TLRs), essential components of the pattern recognition receptor family, are believed to play a vital role in the progression of MS. Elevated expressional levels of TLR2 were observed in peripheral blood lymphocytes and demyelinating regions in CNS of MS patients [27]. Further, elevated levels of soluble TLR2 were indicated in the serum samples obtained from MS patients [28]. The contribution of the TLR2 was also reported in the EAE pathogenesis as an experimental model of MS. In fact, the TLR2-deficient mice display a mild manifestation of EAE [29]. TLR4 was also reported to play a role in the process of MS patients and EAE mice. Its expression level was found to be significantly higher in lymphocytes of MS patients [27]. Some studies also showed that TLR4 deficient (TLR4^−/−^) mice were inadequate to induce EAE through mediating Th17 infiltration in MOG-induced EAE C57BL/6 mice [30]. However, Marta’s paper demonstrated that TLR4^−/−^ mice exhibited more severe EAE symptoms compared to WT mice in the MOG-induced EAE model using C57BL/6 mice [31]. Nevertheless, TLR4^−/−^ mice were also found to show the same susceptibility to disease when compared with WT mice in MOG-induced EAE C57BL/6 mice [29]. Therefore, TLR2 rather than TLR4 might be an important modulator during the process of MOG-induced EAE in C57BL/6 mice, and the elevated protein levels of TLR2 detected in our study coincide with the literature.

In the recent literature, the role of mitochondrial impairment is found to be related to the development of MS. Mitochondria of neurons from patients with MS exhibited abnormalities in mitochondrial protein function, alterations in mitochondrial gene expression levels, and significant deletions of mtDNA, all of which are believed to contribute to various mitochondrial dysfunctions. In active lesions from acute MS patients, there was a global reduction in the total mitochondrial density [32] and a decrease in complex I and III activity [10]. A separate study, which included thirteen individuals with SPMS, discovered substantial mtDNA deletions in neuronal cells, with some cases exhibiting particular deletions in the complex IV subunits [11]. In the EAE model for MS, the mitochondria exhibit morphological changes such as swelling, along with disruptions in mitochondrial function and axonal depolarization [33]. Our data indicated a decrease in protein expression levels of Mitofusion2, OPA1, and Drp1. Mitochondrial membrane proteins OPA1 and Mitofusion2 are essential proteins involved in mitochondrial fusion, while Drp1 plays a crucial regulatory role in mitochondrial fission [34]. Our study revealed that mitochondrial fusion and fission are impaired in EAE mice, while panobinostat is effective in alleviating mitochondrial dysfunction.

Growing research indicates a substantial role of oxidative stress in the development of MS. Reactive oxygen species (ROS)-mediated oxidative stress contributes to MS by acting on distinct pathological processes including blood–brain barrier (BBB) disruption, which in turn promotes leukocyte migration and leads to myelin phagocytosis, oligodendroglial damage, and axonal injury. In MS patients, blood and cerebral spinal fluid (CSF) malondialdehyde (MDA) increased and blood albumin levels decreased, strengthening the clinical evidence of increased oxidative stress in MS [35]. In the EAE model, ROS produced by macrophages may cause mitochondrial dysfunction and localized axonal degeneration [36]. In addition to immunomodulatory and anti-inflammatory treatments, anti-oxidant therapy may offer a potential approach for managing MS. Dimethyl fumarate (DMF), a simple molecule derived from fumaric acid, has received FDA approval as a recommended treatment for MS. DMF was demonstrated to stimulate anti-oxidant response pathways and enhance the expression levels of Nrf2 [37]. NOX2 (Gp91phox) is associated with MS, and its expression is linked to the severity of the condition [38]. Nrf2 has been demonstrated to regulate the expression of enzymes and molecules that are involved in the anti-oxidant defense mechanism. Our data revealed that panobinostat could reduce the expression of NOX2 and HO-1 while enhancing the protein levels of Nrf2, suggesting its potential anti-free radical activity in EAE.

Panobinostat was found to reduce the lymphocyte proliferation response to MOG as well as Th1 and Th17 spinal cord infiltration in a mouse model of PRMS in NOD mice. However, treatment with panobinostat administered after disease onset does not delay the evolution of EAE in NOD mice [39]. In our study, panobinostat was found to alleviate the progression of MOG-triggered experimental autoimmune encephalomyelitis within C57BL/6 mice. MOG-induced experimental autoimmune encephalomyelitis in C57BL/6 mice is a typical animal model of MS to display a chronic-progressive clinical course with T and B cell responses [40,41], while MOG-induced primary progressive EAE in NOD mice has been reported to develop neuroimmunological characterization that mimics progressive MS patients [42]. Nevertheless, the dose-dependent impact of panobinostat needs further clarification. Additionally, it will be interesting to further examine the relationship between the effect of panobinostat on attenuating experimental autoimmune encephalomyelitis and the inhibitory activity of panobinostat on histone deacetylase.

## 4. Materials and Methods

### 4.1. Animal and Drug

Female C57BL/6 mice (16 to 18 g) were procured from Beijing Vital River Experimental Animal Technology Company (Beijing, China). Before commencing the study, the mice were given a week to adjust to the lab environment, which was maintained at a temperature of 24 °C ± 1 °C, a 12 h alternating light and dark cycle, and a relative humidity of 55 ± 5% within the animal facility. The ethical conduct of all animal-based procedures was approved by the Animal Ethics Committee of Chinese Academy of Medical Sciences and Peking Union Medical College (No. 00006750).

### 4.2. Induction of EAE Model

The primary progressive EAE model for C57BL/6 mice was developed using a method outlined in previously published research [25]. The mice were anesthetized and received a subcutaneous injection on the back with 100 μL of 300 μg MOG 35–55 peptide (ChinaPeptides, Suzhou, China) emulsified in complete Freund’s adjuvant (CFA) (Sigma-Aldrich, Saint Louis, MO, USA), which contained *Mycobacterium tuberculosis* H37Ra (Becton, Dickinson and Company, Sparks, NV, USA). Each mouse was injected with a total of 300 μg of MOG 35–55 and 600 μg of *Mycobacterium tuberculosis* H37Ra. Additionally, they received an intraperitoneal (i.p.) injection of pertussis toxin (PTX) (600 ng) (Sigma-Aldrich, Saint Louis, MO, USA) in PBS on the first day of immunization and again two days later.

### 4.3. Panobinostat Administration and Groups

The mice were assigned to three groups in a random manner, with nine animals in each group: control group, EAE group, and EAE + panobinostat group (30 mg·kg^−1^ per day, intragastric administration (i.g.) of panobinostat as an aqueous suspension in 0.5% CMC-Na. Panobinostat was sourced from MedChemExpress (MCE, Shanghai, China). Mice were given panobinostat for a duration of 16 days, while the control and EAE groups were provided with an equivalent volume of 0.5% CMC-Na.

### 4.4. Behavioral Evaluation

Daily observations of the mice were conducted to monitor for disease indicators, which were evaluated based on the 0–5 points clinical scoring criteria [25]. The neuromotor score was evaluated on day 28 post-immunization and the suspension behavior score was assessed on days 7, 14, 21, and 28 post-immunization as described previously [25].

### 4.5. Histology and Immunohistochemistry

The paraffin-embedded lumbar spinal cord sections were stained with LFB to determine the extent of demyelination and with H&E to assess inflammatory infiltration. For immunohistostaining, paraffin sections were treated with primary antibodies against MBP (CST, #78896, Danvers, MA, USA), Iba-1 (CST, #17198, Danvers, MA, USA), and GFAP (Proteintech, #16825-1-AP, Chicago, IL, USA) for evaluating microglial, astrocytes and mature oligodendrocytes, respectively. Images were photographed at a magnification of 100× (HE, LFB, MBP, and GFAP staining) and at a 200× magnification for Iba-1 staining, using the LEICA DM6000B (Leica Microsystems Ltd., Solms, Germany) equipped with a LEICA DFC300 FX (Leica Microsystems Ltd., Solms, Germany). The analysis of the images was performed with the aid of ImageJ 1.52 software, following the previously described methods [25].

### 4.6. Quantitative Real-Time RT-PCR

The spinal cord tissues of mice were harvested. Total RNA was extracted, and quantitative real-time PCR (qRT-PCR) was performed with specific primers and SYBR^®^ Premix Ex TaqTII (TakaRa Clontech, Dalian, China). The primer sequences were following the published primers [25].

### 4.7. Western Blotting (WB) Analysis

For WB, equivalent quantities of spinal cords of mice were employed, and the following primary antibodies were applied: anti-β-actin (Abcam, #ab6276, Cambridge, MA, USA), anti-TLR2 (CST, #13744, Danvers, MA, USA), anti-MyD88 (CST, #4283, Danvers, MA, USA), anti-IRF5 (Santa Cruz, #SC-56714, Santa Cruz, CA, USA), anti-OPA1 (Abcam, #ab157457, Cambridge, MA, USA), anti-Mitofusion2 (Abcam, #ab56889, Cambridge, MA, USA), anti-Drp1 (Abcam, #ab184247, Cambridge, MA, USA), anti-NOX2 (Santa Cruz, #SC-130543, Santa Cruz, CA, USA), anti-Nrf2 (Abcam, #ab62352, Cambridge, MA, USA), and anti-HO-1 (Abcam, #ab13248, Cambridge, MA, USA). Using a Gel-Pro analyzer 4.0 (Media Cybernetics, Bethesda, MD, USA), the band densities were determined.

### 4.8. Statistical Methods

Data were presented as the Mean ± S.E.M. Statistical analysis was performed using either One-way ANOVA followed by Dunnett’s multiple-comparison test or Two-way ANOVA complemented by Sidak’s multiple comparison test. Figure legends specify the number of mice that were used. GraphPad Prism Version 7.0 software (GraphPad Software Inc., La Jolla, CA, USA) was used for all statistical analyses. A *p*-value below 0.05 was established as the criterion for statistical significance.

## 5. Conclusions

In conclusion, our study first identified the potential activity of panobinostat on MOG-induced EAE in C57BL/6 mice. The study provides a potential compound for the treatment of MS and further demonstrates the feasibility of applying HDACis in MS.

## Figures and Tables

**Figure 1 ijms-25-12035-f001:**
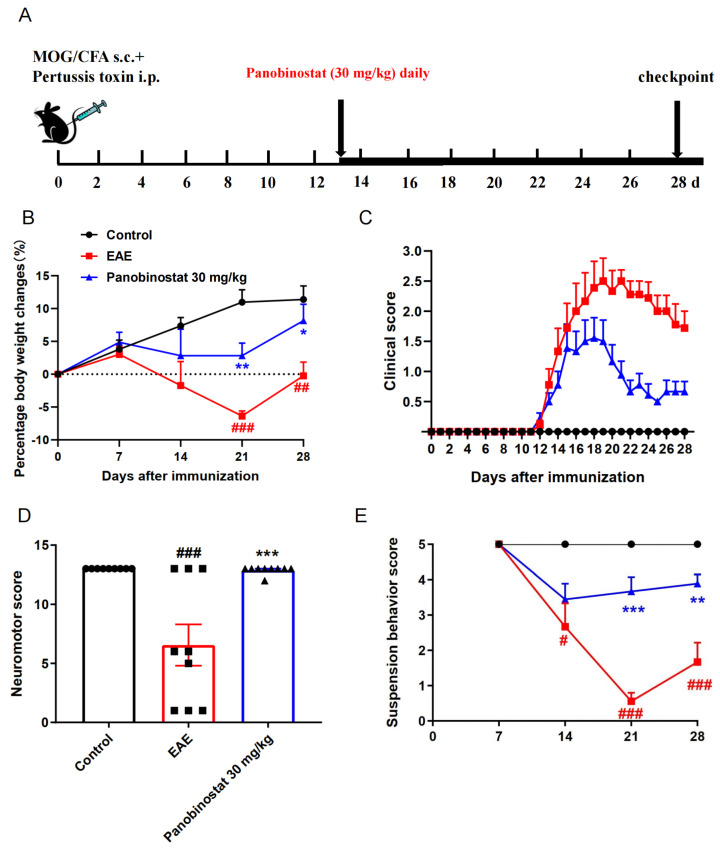
Panobinostat ameliorates the symptoms of EAE mice. (**A**) The process of this experiment. (**B**) Time course percentage changes in body weight in the mice from the respective group. (**C**) Temporal changes in the clinical scores for mice in each group. (**D**) The average neuromotor scores across the different groups. (**E**) The suspension behavior score was recorded on days 7, 14, 21, and 28 post-immunization in each group. Data are depicted as mean ± SEM (*n* = 9). ^#^ *p* < 0.05, ^##^ *p* < 0.01, ^###^ *p* < 0.001 vs. the control group; * *p* < 0.05, ** *p* < 0.01, *** *p* < 0.001 vs. the group with EAE.

**Figure 2 ijms-25-12035-f002:**
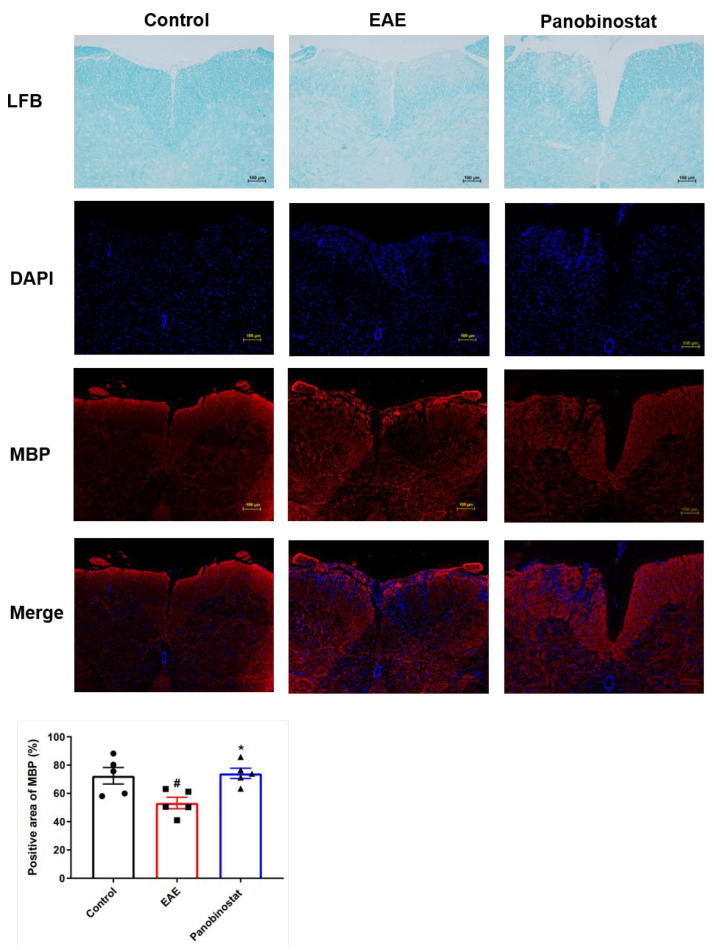
Panobinostat averted the demyelination and depletion of oligodendrocytes located within the spinal cord tissues of mice affected by EAE. On day 28 post-immunization, spinal cord sections were stained with MBP to identify mature oligodendrocytes. Data are expressed as the mean with SEM indicated (*n* = 5), with a significant difference denoted by ^#^ *p* < 0.05 compared to the control group; and * *p* < 0.05 denoting significance relative to the group with EAE.

**Figure 3 ijms-25-12035-f003:**
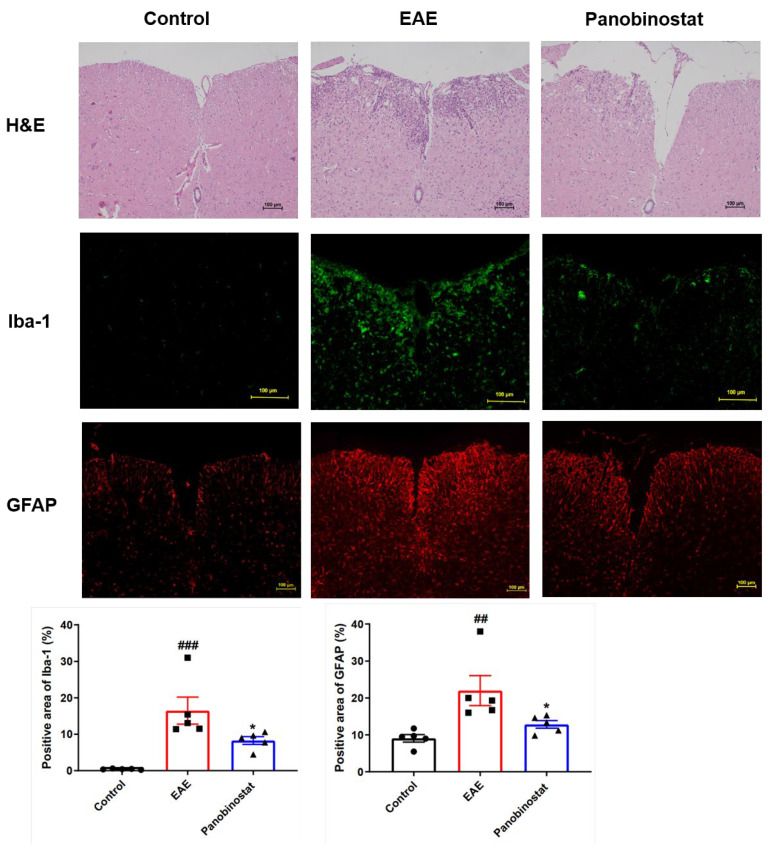
Panobinostat attenuated the CNS inflammation and the microgial/astrogial activation in the spinal cord tissues of mice with EAE. Sections of the spinal cord were subjected to histological staining with H&E, Iba-1 for microglia, and GFAP for astrocytes on day 28 post-immunization. Values are presented as mean ± SEM (*n* = 5), ^##^ *p* < 0.01, ^###^ *p* < 0.001 vs. the control group; * *p* < 0.05 vs. the group with EAE.

**Figure 4 ijms-25-12035-f004:**
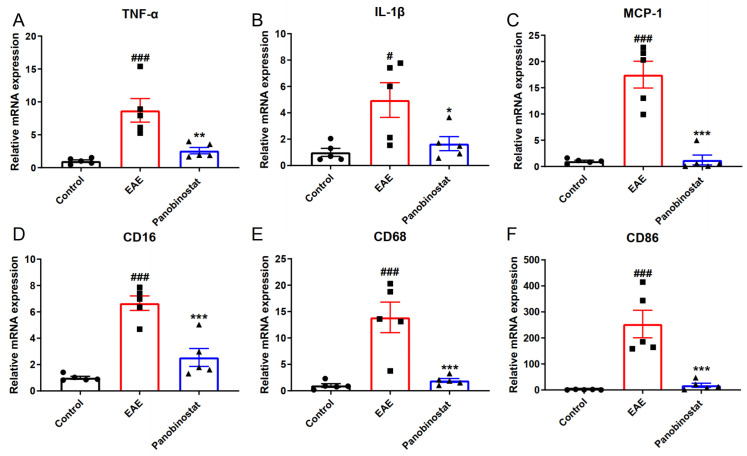
Panobinostat suppressed the polarization of M1 microglia and the generation of pro-inflammatory factors in the spinal cord tissues of mice with EAE. The mRNA levels of pro-inflammation factors (TNF-α, IL-1β, and MCP-1) (**A**–**C**) and M1 microglial markers (CD16, CD68, and CD86) (**D**–**F**) as ascertained by quantitative real-time RT-PCR. Results are shown as the mean ± SEM (*n* = 5), ^#^ *p* < 0.05, ^###^ *p* < 0.001 vs. the control group; * *p* < 0.05, ** *p* < 0.01, *** *p* < 0.001 vs. the EAE group.

**Figure 5 ijms-25-12035-f005:**
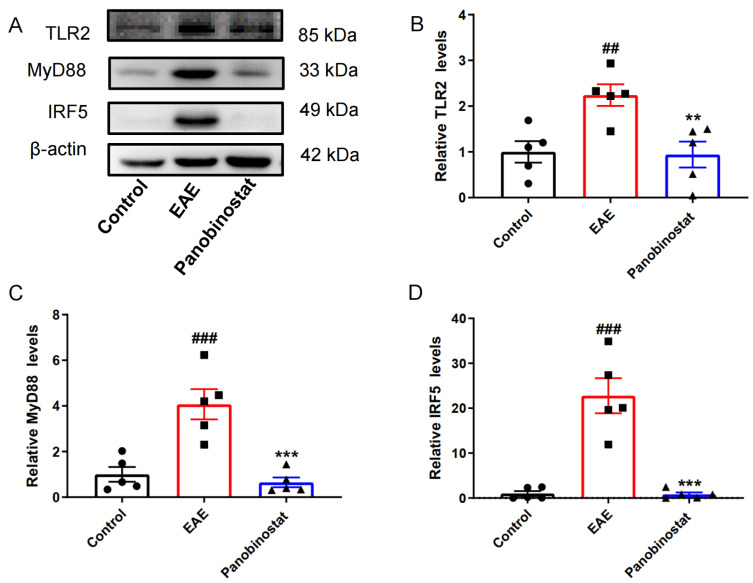
Panobinostat reduced the levels of TLR2/MyD88/IRF5 expression in the spinal cord tissues of mice with EAE. The representative blots (**A**) and the quantitative analysis of protein levels (**B**–**D**). Results are shown as the mean ± SEM (*n* = 5), ^##^ *p* < 0.01, ^###^ *p* < 0.001 vs. the control group; ** *p* < 0.01, *** *p* < 0.001 vs. the EAE group.

**Figure 6 ijms-25-12035-f006:**
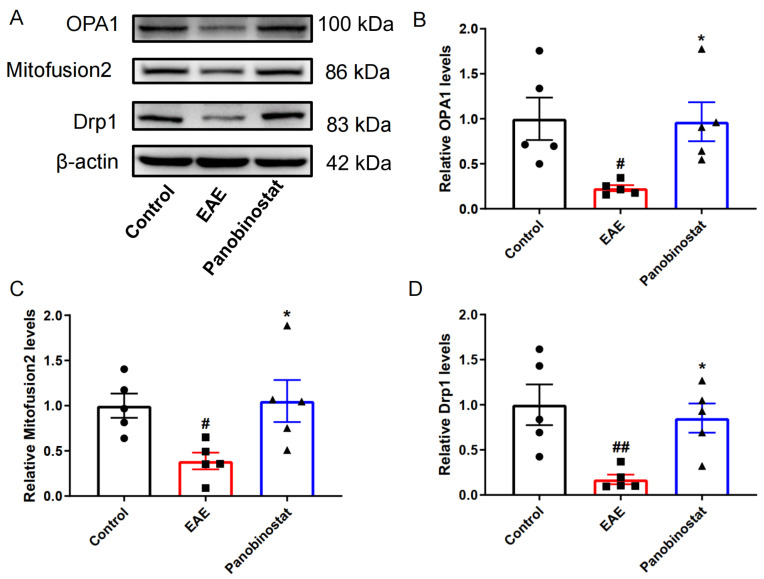
Panobinostat attenuated the dysfunction associated with mitochondria in the spinal cord tissues of mice with EAE. (**A**) Representative blots. (**B**–**D**) Quantitative analysis of protein levels presented in (**A**). Results are shown as the mean ± SEM (*n* = 5), ^#^ *p* < 0.05, ^##^ *p* < 0.01 vs. the control group; * *p* < 0.05 vs. the EAE group.

**Figure 7 ijms-25-12035-f007:**
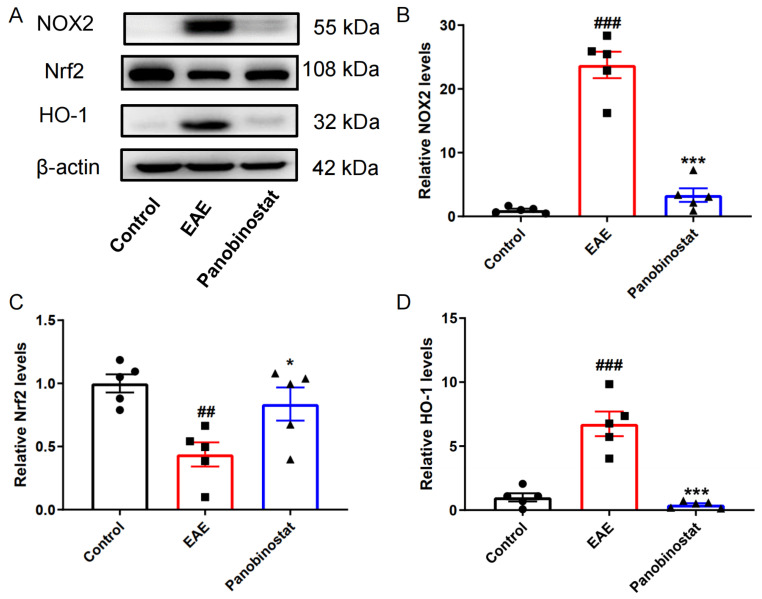
Panobinostat reduced oxidative stress in the spinal cord tissues of mice with EAE. (**A**) Representative blots. (**B**–**D**) Quantitative analysis of protein levels shown in (**A**). Results are shown as the mean ± SEM (*n* = 5), ^##^ *p* < 0.01, ^###^ *p* < 0.001 vs. the control group; * *p* < 0.05, *** *p* < 0.001 vs. the EAE group.

## Data Availability

The data generated and analyzed in this paper are available from the corresponding author upon reasonable request.

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
