# Peer review of "Panobinostat Attenuates Experimental Autoimmune Encephalomyelitis in Mice via Suppressing Oxidative Stress-Related Neuroinflammation and Mitochondrial Dysfunction"

_ijms, 2024, doi:10.3390/ijms252212035_

Round 1

Reviewer 1 Report

Comments and Suggestions for Authors

Although several anti-inflammatory drugs have been discovered or developed against autoimmune diseases, the solely reducing the effect is via disease-modifying treatment rather than disease-specific treatments.

Despite well-known immunomodulators, targeting specific niches where the control point of the autoimmune diseases is most important. Shen et al. have submitted the manuscript entitled “Panobinostat attenuates experimental autoimmune encephalomyelitis in mice via suppressing oxidative stress related neuroinflammation and mitochondrial dysfunction”.

This reviewer has the following suggestions:

Figure 1: Please illustrate the experimental design, which is essential for understanding the experimental setup.

Figure 1A: Please present the percentage body weight changes rather than body weight changes.

Figure 1C: SEM is missing in the control group.

All bar graphs in the MS should be represented in scattered form; as n=5 mice.

Regardless of the PCR results, use the flow cytometry method to distinguish M1 over M2.

Why didn't the authors use ELISA to assess the cytokine expression in spinal fluid?

Section 4.2: “Immunized on the back” do you mean subcutaneously?
Why the pertussis toxin and in two doses used in the procedure, this should be explained.

Why the CMC-Na used as a vehicle control. Did the authors use the same formulation for intragastric administration? This should be clearly explained. 

Reviewer 2 Report

Comments and Suggestions for Authors

This manuscript deals with the ability of the panobinostat to attenuate the experimental autoimmune encephalomyelitis (EAE) in mice.

It appears that panobinostat can affect the oxidative stress associated with neuroinflammation and alterations in mitochondrial functionality.

Panobinostat is an HDAC inhibitor, in addition to inhibiting other deacetylases. It influences the function of cytochromes, and thus it is not surprising it can affect mitochondrial function.

From this work, it is not clear whether it inhibits HDAC or other deacetylases.

The data shown are clearly reminiscent of what was described by the authors (see reference 25)  for another HDAC inhibitor.

The images of histological specimens should be enlarged, and a higher magnification of the immunofluorescence assay should be shown.

The beta-actin used as a reference for the western bolt appears to be quantitatively different in different gels under the same experimental conditions. This would suggest that beta-actin can vary among samples under the same experimental conditions. Another control should be used.

The effect of the inhibitor is evident only if the treatment starts at the same time as the induction of EAE. This would suggest that the inhibitor is just blocking the initial inflammation and not the evolution of the EAE.

This lack of effect would suggest that the inhibitor can play a role simply because it affects proliferation (it is used as an antineoplastic agent in Multiple Myeloma patients). It is not clear if this is linked to HDAC activity.

Finally, how much EAE in mice can indeed resemble human MS?    

Comments on the Quality of English Language

English is good enough.

Round 2

Reviewer 1 Report

Comments and Suggestions for Authors

The authors have revised the manuscript by considering the reviewers' suggestions. The revised manuscript is acceptable for publication in its current form. I here by endorse the manuscript for publication. 

Reviewer 2 Report

Comments and Suggestions for Authors

The authors revised the manuscript according to reviewers' suggestions.

Comments on the Quality of English Language

The English language is good enough.